# The School Attachment Monitor—A novel computational tool for assessment of attachment in middle childhood

**Maki Rooksby[1], Simona Di Folco[2], Mohammad Tayarani[3], Dong-Bach Vo[3], Rui Huan[3], Alessandro Vinciarelli[3‡], Stephen A. Brewster[3‡], Helen Minnis[1‡]***

1 University of Glasgow, Institute of Health and Wellbeing, Glasgow, United Kingdom, 2 University of Edinburgh, School of Health in Social Science, Edinburgh, United Kingdom, 3 School of Computing Science, University of Glasgow, Glasgow, United Kingdom

‡ These authors are joint senior authors on this work.
* helen.minnis@glasgow.ac.uk

## Abstract

**Data Availability Statement:** Because the data cannot be fully de-identified (since they include video data of children) we cannot make them available to other researchers as this would breach

### Background

Attachment research has been limited by the lack of quick and easy measures. We report development and validation of the School Attachment Monitor (SAM), a novel measure for largescale assessment of attachment in children aged 5–9, in the general population. SAM offers automatic presentation, on computer, of story-stems based on the Manchester Child Attachment Story Task (MCAST), without the need for trained administrators. SAM is delivered by novel software which interacts with child participants, starting with warm-up activities to familiarise them with the task. Children's story completion is video recorded and augmented by 'smart dolls' that the child can hold and manipulate, with movement sensors for data collection. The design of SAM was informed by children of users' age range to establish their task understanding and incorporate their innovative ideas for improving SAM software.

### Methods

130 5–9 year old children were recruited from mainstream primary schools. In Phase 1, sixty-one children completed both SAM and MCAST. Inter-rater reliability and rating concordance was compared between SAM and MCAST. In Phase 2, a further 44 children completed SAM complete and, including those children completing SAM in Phase 1 (total n = 105), a machine learning algorithm was developed using a "majority vote" procedure where, for each child, 500 non-overlapping video frames contribute to the decision.

### Results

Using manual rating, SAM-MCAST concordance was excellent (89% secure versus insecure; 97% organised versus disorganised; 86% four-way). Comparison of human ratings of SAM versus the machine learning algorithm showed over 80% concordance.

the ethical approval under which these data were gathered (University of Glasgow College of Science and Engineering Ethics Committee, application number 300140024). The person to contact for enquires about the data is Dr Christoph Scheepers Christoph.Scheepers@glasgow.ac.uk, Chair of the Ethics Committee that approved the study.

**Funding:** Engineering and Physical Sciences grant EP/M025055/1 to SB, AV and HM EPSRC.UKRI.org The funders had no role in study design, data collection and analysis, decision to publish, or preparation of the manuscript.

**Competing interests:** The authors have declared that no competing interests exist.

## Conclusions

We have developed a new tool for measuring attachment at the population level, which has good reliability compared to a validated attachment measure and has the potential for automatic rating–opening the door to measurement of attachment in large populations.

## Introduction

Attachment is a fundamental element of human psychological development and has been examined in many thousands of studies worldwide [1]. However, the attachment measures with good psychometric properties [1, 2] and a history of use in multiple studies, for example the Strange Situation (SSP) [3, 4], and the Manchester Child Attachment Story Task (MCAST) [5], were developed for research purposes. Intensive training is required to achieve reliability and these "gold standard" measures are time-consuming to administer and rate [6, 7]. This has prevented attachment research fully influencing clinical practice [8]. We aim, in this study, to develop and test an attachment measure that is quicker and easier to administer and rate than the existing measurement tools.

Attachment measures such as SSP and MCAST provide a standardised, replicable protocol that presents the child with a series of stressful scenarios in the presence of the actual parent/caregiver (SSP) or a representation of the child and parent/caregiver (e.g. using drawings or doll play–MCAST). Classification of attachment is achieved by direct observation of actual or represented child and parent/caregiver behaviours, and rigorous training is required to ensure these observations are valid and reliable [8]. The Strange Situation Procedure (SSP) [9] is most valid when used when the child is aged 12 to 18 months and can also be used in the pre-school years [8]. The Manchester Child Attachment story task–most valid when used in children aged 4 to 9 years of age–has been used in more than 25 studies across 9 countries and has had rigorous psychometric testing with adequate inter-rater [10], test-retest reliability and criterion validity [8].

Administering the SSP in clinical practice takes approximately one hour of two staff members' time: a specifically trained administrator and a "stranger" plus at least an additional hour of rating time. Training to administer and rate the SSP requires a two-week (usually US-based) course, then rigorous reliability training to reach at least 80% agreement with expert raters. Thereafter, high quality research labs achieve four-way attachment classifications of variable reliability (Cohen's Kappas 0.49 to 0.93) [11, 12]. Therefore, despite the recommendation for these attachment measures (e.g. by the UK National Institute for Health and Care Excellence [13]), their use is rarely feasible in clinical practice.

Despite the large research database on attachment that has accrued since the 1970s, important questions remain unanswered. These include how genetics and environment interact in the development of attachment relationships across the lifespan [14] and the potential that secure attachment could act as a "susceptibility factor", encouraging children to explore the environment, resulting in either positive or negative outcomes depending on the risks and protective factors they encounter [15]. To fully examine these important but complex questions, much larger samples will be necessary—in the order of the tens of thousands of participants [16].

Over a decade ago, we set out to address the challenge of finding a quicker and easier, yet valid, way to assess attachment patterns in early to middle childhood. First, we found that no quick and easy childhood measures of attachment existed [17], as two more recent systematic

reviews have confirmed [8, 18]. Rapid and easy measures of attachment patterns in young children do not exist for good reasons: attachment is a relational process involving measurement of the child's attachment behaviours, the caregiver's caregiving behaviours and the degree to which the child's distress is assuaged [10]. Because attachment behaviours are only activated when the child is stressed by fear, hunger, illness or other aversive situations, these attachment and caregiving behaviours can *only* be observed in a stressful situation [19]. Questionnaire or interview measures cannot, therefore, directly measure attachment, especially in early childhood when children do not have adequate cognitive and language abilities to allow meaningful self- report [8]. In developing our new tool, we focused on early to middle childhood when interventions can be highly effective [20] and most children are in school, allowing widespread data collection for public health or epidemiological purposes. We developed a "computer game" version of the Manchester Child Attachment Story Task, called CMCAST [7], in which the child interacted with a dolls-house as depicted on a laptop computer screen and an avatar of the child-doll and mummy-doll which he/she could move around on the screen to complete the story. The rater could then view a short animation of each story and see a cut-in video of the child as s/he told the story (see S1 Fig).

We compared the CMCAST with the doll-play original MCAST by randomly allocating the order in which children to received CMCAST and MCAST six weeks apart, and also conducted a feasibility study on the self-administration of the CMCAST in small groups in school classrooms [7]. We demonstrated that the CMCAST could be administered to school-age children with minimal adult supervision and in a classroom setting with groups of up to five children simultaneously. We were forced to pare down the CMCAST rating system to five simple elements: *child engagement* (i.e. was there a visible change in child facial expression at the crisis point of the story stem?); *child doll attachment behaviours* including proximity-seeking between child doll and mummy doll; *caregiving behaviours* from the mummy doll as depicted in the child's story; *resolution of the attachment stress* as depicted in either the child's story or during the post-story prompts and finally *exploratory behaviour* as depicted in the child's story. This was a much simplified rating system compared to the more than 20 items required to classify attachment and associated features in the MCAST. Despite this, our study resulted in an almost identical distribution of attachment classifications and was as reliable as the MCAST doll-play original [7]. An independent study has since rated MCAST and CMCAST positively in terms of content validity with respect to attachment theory, and commented on the potential advantages of the simpler rating system for CMCAST [8].

The efficiencies and cost savings associated with CMCAST were, however, still not enough of an advance over the established tools to justify production of the game. CMCAST raters still needed to attend several days of training in administering and rating the MCAST and had to work hard over several months to achieve rating reliability of both the MCAST and CMCAST. It was therefore clear that if a computerised easy-to-administer attachment measure were to be scalable and have a useful impact on attachment research and clinical practice, then automatic rating would be the next important step forward.

Simultaneously with the development of the CMCAST, we were developing an observational tool to help clinicians observe indiscriminate behaviours in the clinic waiting room. Findings showed that in an unfamiliar and therefore stressful clinic setting, typically developing children stayed very close to their parents [21] seeking for or maintaining proximity. Observing young children with their parents in SSP and in clinic waiting rooms, and observing doll behaviour in the doll-play MCAST or on the computer screen, secure attachment seemed to be characterised by *smooth* and fairly rapid movement to reduce the distance between child and mummy (doll). Movement is now recognised as a key element of human interaction and an important substrate of communicative development from before birth [22]. Problems in

the subtleties of body movement, such as in autism, can impede social interaction [23]. The importance of movement in attachment patterns has been touched on in psychotherapeutic research [24] and practice [25], but has been barely investigated in early/middle childhood attachment research [26]. Yet movement was a core aspect of Bowlby's description of attachment behaviour: in 1958, he described the instinctual nature of the proximity-seeking seen in both infants and frightened older children, emphasising that this "following behaviour" is seen across many species [27]. In a response to Bowlby's paper Schaffer and Emerson suggested that proximity-seeking was "the core of the attachment function" and was "a relatively clear-cut, easily identifiable behavior tendency which may be observed to occur almost universally amongst animals as well as in man"(page 6) [28]. Bowlby cautioned that, although the centrality of movement was evident across species, there might also be "species-specific signifiers" in the way particular animals exhibit attachment behaviour. In humans, language could be one of these, but Bowlby suggested that "beneath the symbolic transformations and trappings of humanity, [there are] primeval dynamic structures which we share with lower species" (page 365) [27]. Despite Bowlby articulating the need for research on these aspects, such research has never been conducted.

In 2009, Vinciarelli and colleagues launched the new computer science of "Social Signal Processing", asserting that the understanding of the social signals that facilitate human communication is a crucial aspect of human intelligence, and describing machine detection and interpretation of these behavioural cues [29]). Since then technological advances in computing science, including the development of small sensors [30]) and machine learning algorithms for analysing human interactions [31], have allowed this field to move forward significantly.

We have now engaged some of the new advances in social signal processing in an attempt to develop a new computer "game" for measuring attachment, including movement sensors embedded in dolls and machine learning algorithms with a view to an automatic rating system [32]. We have called this new computerised version of the Manchester Child Attachment Story Task (MCAST) the School Attachment Monitor (SAM), because we hope it can eventually become an efficient, low-cost way of measuring attachment in young school-age children that can be used in school classrooms for screening/mapping purposes or for data-gathering in epidemiological research.

**Our research questions were**

1. Can acceptable inter-rater reliability be achieved for manual rating of SAM?

2. Is there good agreement on two-way and four-way attachment classifications between SAM and the MCAST doll-play original when manually rated by raters trained to reliability on MCAST rating?

3. For SAM, is there good agreement between a machine learning algorithm and the manual classification of secure versus insecure attachment?

## Methods

The study received ethical approcal from the University of Glasgow College of Science and Engineering Ethics Committee and the governing bodies of the local councils responsible for participating schools.

## Development of SAM

SAM was developed with close reference to the development of the Computerised MCAST, learning from design mistakes/successes with that instrument [7]. Simplicity in design (a

dolls-house with four rooms, no stairs and only furniture necessary for the stories and simple dolls with humanoid characteristics but no attempt at realism) aimed to a. avoid over-stimulating the child, b. facilitate the projective identification of the child with the child doll and c. encourage translation of the child's inner world into the play.

## Stakeholder workshop 1

In order to make small changes in the presentation of SAM compared to CMCAST, a design process workshop was held with 13 typically developing children aged between 5 and 10 (7 boys; 6 girls) recruited through the personal network of the authors. The aim was to test whether children were able to engage in representational attachment story-telling on a two-dimensional surface as effectively as the traditional three-dimensional MCAST doll's house. Children responded to up to 2 vignettes each from the MCAST using either the traditional doll's house or using the dolls on a two-dimensional fuzzy felt mat as shown in Fig 1A. The order of version presentation was counterbalanced so that approximately half the children experienced doll's house version first and vice versa. Their responses were video recorded. Children of this age were perfectly capable of a) understanding and using a two dimensional plan version of the dolls-house in which a simple fuzzy felt mat was divided by a fuzzy felt cross into four rooms and b) of having a story stem told to them on the screen, then completing the story on the dolls-house mat, rather than on the computer screen as demonstrated by their engagement in the task and production of stories of a rateable quality.

The use of the two-dimensional fuzzy felt mat for the dolls-house allowed us to embed tiny movement sensors into actual dolls (see Fig 1B). SAM was therefore more like the MCAST original than the CMCAST had been in that children could manipulate both "mummy doll" and "child doll" simultaneously using real dolls (rather than screen dolls), could exploit the full

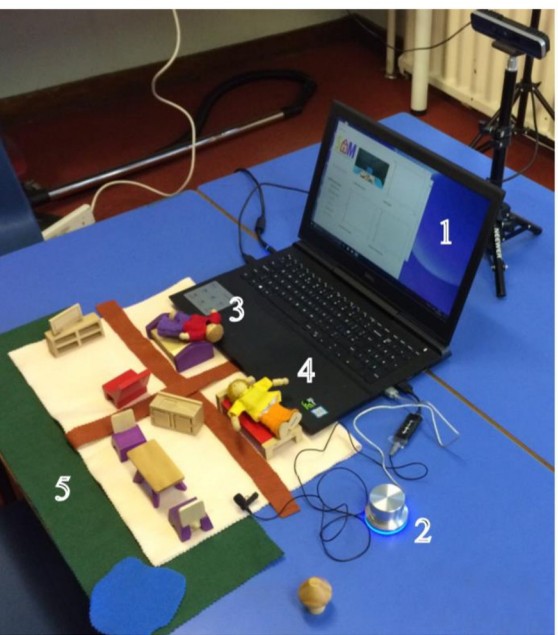
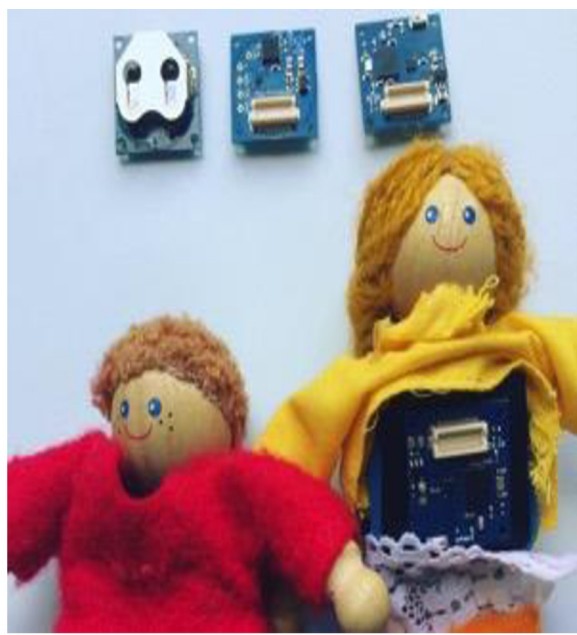

**Fig 1. The components of SAM. a.** (above left): shows the system as it appears to the users. The computer screen (**1** in the picture) displays the videos with actors guiding the child through the MCAST, the button (**2**) allows the users to signal the system that they have completed an MCAST step, the dolls (**3** and **4**) and the toy house (**5**) allow the users to complete the story stems. **b.** (above right): An image of the dolls that we have designed and installed with sensors inside to enable administration and machine-learning based ratings.

range of three dimensional doll movements and, like in MCAST, could fully express themselves. Like CMCAST, the entire doll and child action was videotaped with the child's face and hands in the frame. Our administrating laptops were equipped with a specialised video camera which recorded children's facial and upper body image as well as tracking depth to allow calculation of the distance between the dolls during their story completion. From the user perspective, these modifications simplified SAM set-up compared to CMCAST since story stems could potentially be delivered on any school computer (laptop or PC) installed with our SAM software, and the only other specific "kit" required would be the dolls-house mat, furniture, computerised dolls and the webcam-like camera with depth sensor. These are light and highly portable compared to the MCAST original set-up.

In order to orientate the child user towards interacting with the computer in SAM, a cat character/avatar greets the child, asks the child to position themselves so that they are captured well by the camera of the SAM set-up in a "posing for a photo" game, explains to the child what their task is, and introduces the vignettes. The five MCAST story stems (i.e., Breakfast, Nightmare, Hurt knee, Illness and shopping) are presented via a recorded performance by a professional actor who acts the beginning of the vignettes using a doll's house. For both MCAST and SAM, the Breakfast vignette is a warm-up task which does not contribute to attachment classification. Further details are available in Vo et al. [32].

### Stakeholder workshop 2

After the *Phase 1* data collection (see below and Fig 2) a second design workshop, with the same 13 children, aimed to address system errors that had been encountered during Phase 1. These were simulated and embedded in a live puppet show with one of the researchers acting as "a puppet who needs help while playing the SAM computer game". Children were asked to imagine a solution so that the puppet would not come across the problem again and to create a soft dough model or drawings to represent their ideas. Solutions offered included a push button with multi-functionality to communicate with SAM (see Fig 1A, element 2). In the final SAM model, children pressed this button to "talk" to the system at various points, e.g. when they have finished telling their story and are ready to move on, affording the child user a sense of agency and autonomy, essential for optimal attachment measurement in this age group. Further details are reported in [33] and the final set up is shown in Fig 1A and 1B [32].

### Testing of SAM

Testing of SAM was conducted in two phases: *Phase 1* in which manual SAM ratings were compared with manual MCAST ratings and *Phase 2* in which an automatic SAM rating system was developed and tested against the manual SAM ratings.

### Data collection for Phases 1 and 2

We approached schools in local councils in which the demographic background included wealthier and more materially deprived and both urban and rural areas of the West of Scotland. The Scottish Index of Multiple Deprivation (SIMD) decile, for the school addresses, ranged between 4 and 9 where the lower the value, the higher the level of deprivation for the area.

We worked with relevant staff members at each participating school to obtain consent from parents and carers of children in eligible age range, which corresponded to Years 1 to 4 (P1-4) in the mainstream primary education under the UK system. Information packs were given to the children by the school offices to take home to pass onto their families, as part of an established system for sending letters and homework to and from homes. Our information sheet explained the main aims and requirements of the study for adults (parents and carers) as well

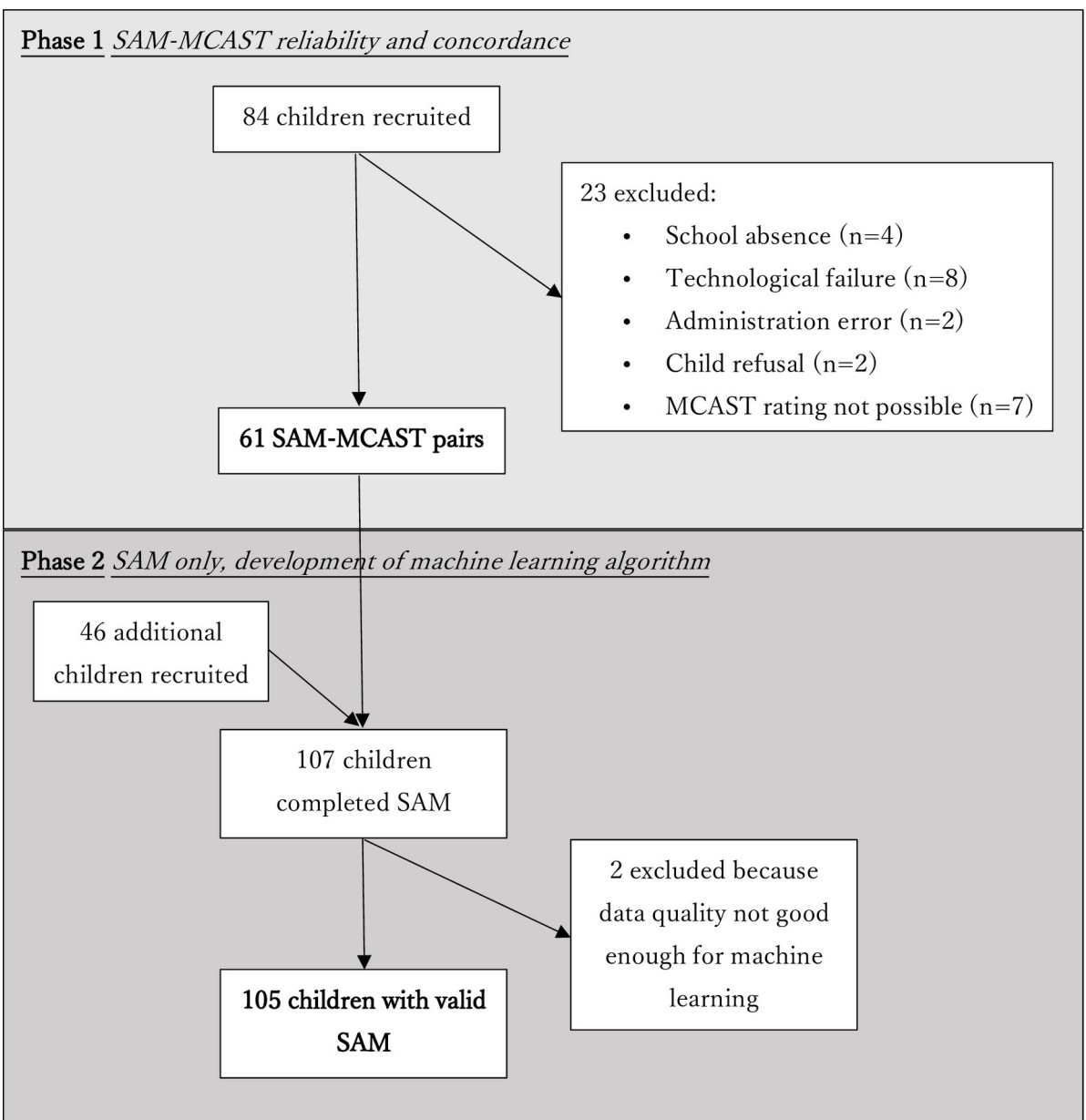

**Fig 2. Data collection Phases 1 and 2.**

as in simple language with pictures for children, so that families could discuss participation with their children at home. Questions regarding participation and any concerns were addressed directly to the research team and families willing for their child to participate were asked to return signed consent forms by a specified date. Consent rates across the 5 schools ranged between 30 to 40%. Only children with signed parental consent were included in the study.

For Phases 1 and 2,130 children aged 5–9 (67; 51% girls) were recruited from the first 4 year groups of 5 mainstream primary schools in 3 local councils (Fig 2). For data security reasons, schools were unwilling to give exact dates of birth of the children but the distribution across school grades and age range/gender balance for each grade are summarized in Table 1.

**Table 1. School grade, gender and age spread (%) of participating children.**

| School Grade | Primary 1 (age 5–6) | Primary 2 (age 6–7) | Primary 3 (age 7–8) | Primary 4 (age 8–9) | Total |
|---|---|---|---|---|---|
| **Number of children (%)** | 28 (21.5%) | 46 (35.4%) | 35(26.9%) | 21 (16.2%) | 130 |
| **% female** | 43% | 52% | 49% | 57% | 51% |

## Phase 1 –SAM–MCAST reliability and concordance

This phase addressed research Questions 1 and 2:

1. Can acceptable inter-rater reliability be achieved for manual rating of SAM?

2. Is there good agreement on two-way and four-way attachment classifications between SAM and the MCAST doll-play original when manually rated by trained raters?

Given our previous success with reliability and concordance comparing the CMCAST to the MCAST [7], we predicted that the new SAM system would produce data of good enough quality to allow reliable rating by trained assessors. Further, we hypothesised that the ratings across the two measures (MCAST and SAM) would show an acceptable concordance with each other as regards secure versus insecure classifications and organised versus disorganised classifications.

Each child was asked to use SAM or MCAST at least six weeks apart and data collection was organized so that similar numbers of children used MCAST first or SAM first. In Phase 1, 84 children were recruited and 61 had rateable data on both MCAST and SAM (see Fig 2).

## Manual rating

Three raters conducted the initial manual ratings of all 61 SAM and 61 MCAST cases. One of the raters (MR) acted as a coordinator for assigning cases between the 3 raters to minimize rating bias, taking care to avoid any rating of sessions where the rater had also acted as the administrator.

As described in Fig 3, for training and quality control purposes, all 61 Phase 1 SAM cases were independently double-rated by at least two raters, then if necessary discussed in consultation with a third expert rater to agree a shared rating. Twenty percent (i.e. n = 12) of the 61 MCAST cases were independently re-rated by our expert rater (SDF). After the initial training period, SDF was consulted if a rater encountered difficulties in making a judgement, if cases showed signs of attachment disorganisation or where data quality raised a question as to whether the case was rateable.

After first rating, 11 (18%) of SAM—MCAST pairs showed significant discordance or were still challenging to rate. Since these ratings had been conducted while the team of raters was inexperienced, SDF conducted an independent re-rating of all 11 SAMs and all 11 MCASTs, never rating the SAM/MCAST for the same child on the same day. All of these ratings were then conferenced with another expert rater, HM. In order to ensure that these second ratings were unbiased by any previous MCAST or SAM rating, details of initial SAM and MCAST ratings were stored separately in a folder that could not be accessed by HM or SDF, and at no time was the SAM and the MCAST of any child discussed simultaneously during the same conference. During these conferences, each of three raters (HM, MR and SDF) examined each video and made an independent rating which was then discussed, referencing the MCAST rating manual, version 26, throughout.

On average each rater was able to rate each case within 1.5 to 2 hours for MCAST, after the initial training period, and within an hour for each SAM case. We estimate that the entire

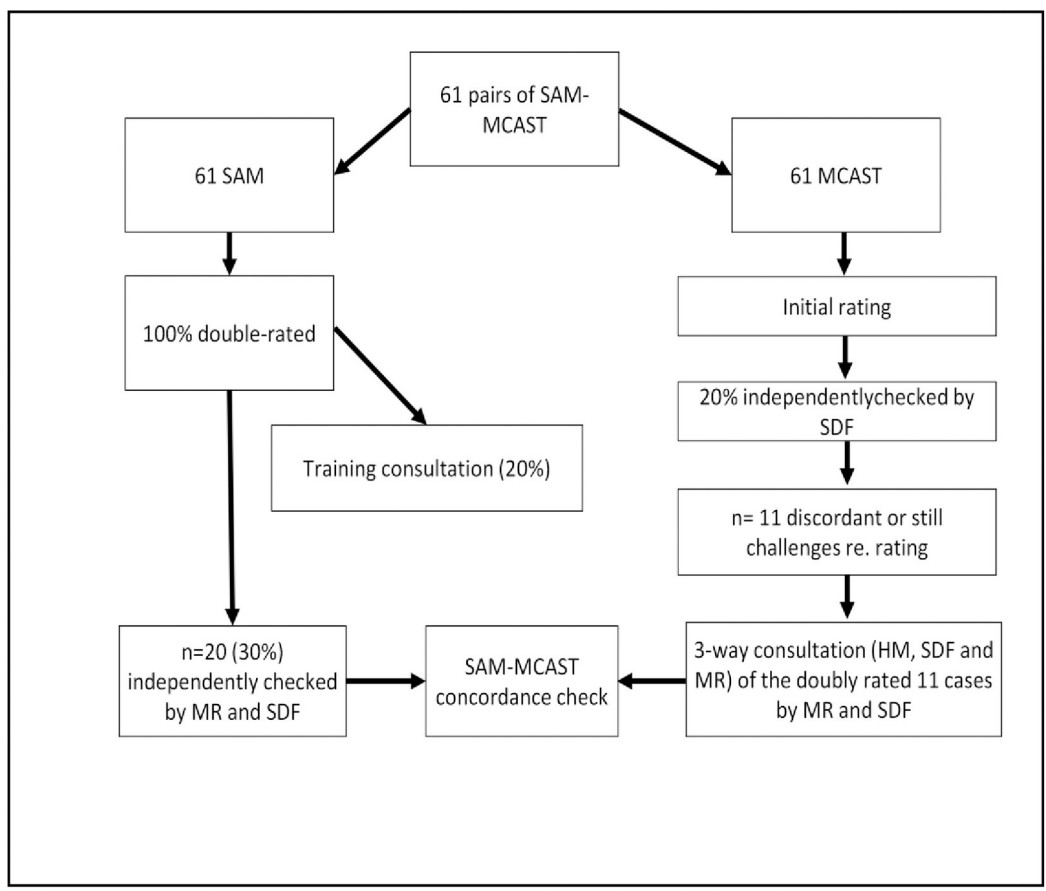

**Fig 3. Phase 1 rating and resolution process including role of expert rater.**

manual rating process, including individual ratings, second independent ratings and conferences, amounted to nearly 500 person-hours.

### Phase 2 –development of machine learning algorithm for SAM

This phase addressed research Questions 3:

3. For SAM, is there good agreement between a machine learning algorithm and the manual classification of secure versus insecure attachment?

Having established the SAM-MCAST concordance in Phase 1, an additional 46 cases of SAM data were collected in order to have a sufficient sample size for the development of the machine-learning (ML) algorithm. Two of these 46 cases were unfit for ML analysis due to data quality. Together with the 61 SAM cases from Phase 1, the ML development reported here was based on a total of 105 sets of SAM data (see Fig 2). In parallel, and independently from, the manual ratings, automated ratings were made as described in more detail in the S1 Appendix and in Roffo et al. [34]. In brief, the proposed attachment recognition approach (full description available in Roffo et al., 2019) [34] started by extracting the pose of the children from every frame of the videos recorded during the MCAST administration. This task was performed with OpenPose, a publicly available software widely applied in the computing community [35].

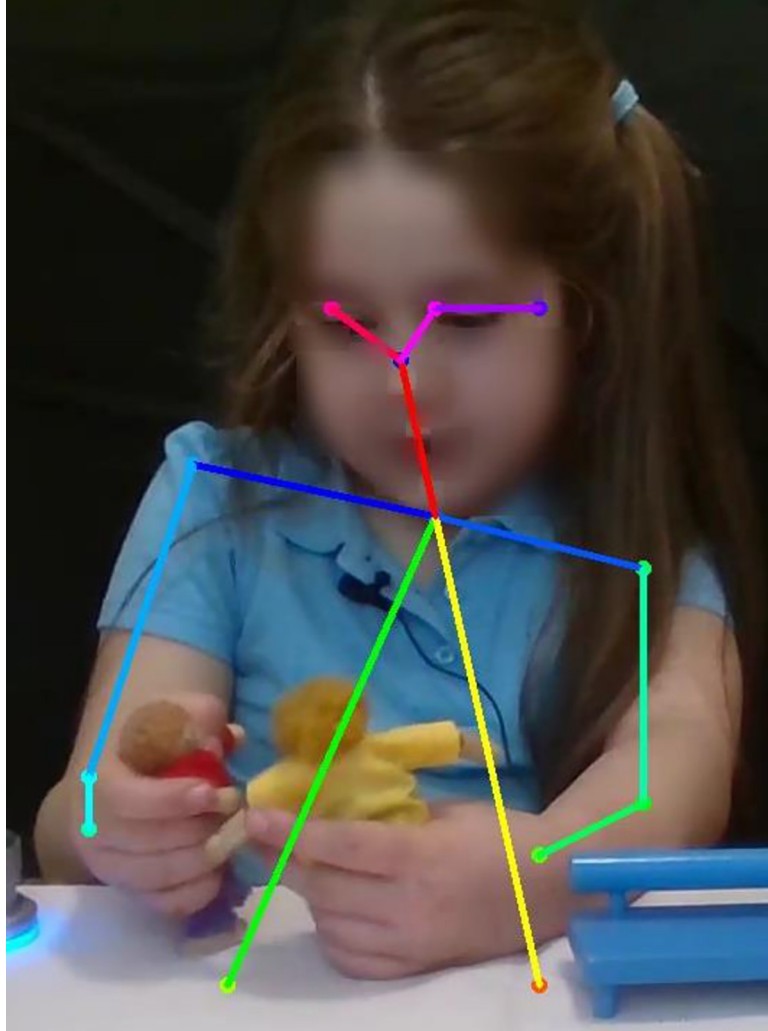

**Fig 4. Production of a generic frame in a video using OpenPose software.**

The pose corresponds to the position of all joints visible in the frames (e.g., wrists, elbows, etc.) and it is represented as a vector where every component is a physical measurement that captures a salient aspect (e.g., the distance between the hands, the speed of hand movements, etc.). After such a process, each video is converted into a sequence of feature vectors that can be modelled with Long Short Term Memory Networks [36], one of the most common models for the classification of sequential data. Given the computational issues resulting from training such networks, the sequence of feature vectors is segmented into non-overlapping windows that account for roughly 4.25 seconds (every window corresponds to 128 feature vectors). Each window is classified separately as corresponding to a secure or insecure child. All decisions made at the level of individual windows are then aggregated through a majority vote, meaning that children are assigned to the class their segments are most frequently assigned to (See Fig 4). For SAM, the focus was on the position of the hands because of our hypothesis that the way the participants move the dolls, held in their hands, to complete the story stems is crucial for classification. We therefore focussed particularly on the following features: *Hand position; Distance between the hands; Hand speed; Hand acceleration; Hand 1D trajectory* (i.e. placing of the hands with respect to the width of the video image); *Hand presence* (i.e. the

proportion of frames in which each of the two hands is present, giving information on the child's tendency to move one doll more than the other).

## Used with permission from Roffo et al., 2019 and with parental consent for the use of the image

The resulting preliminary algorithm was trained using the "Leave-One-Child-Out" protocol i.e. by training the algorithm with all children except one and then testing it with the only child left out. At each iteration, a different child is left out for testing and the process is therefore repeated 105 times. This allows use of the entire dataset while ensuring rigorous independence between the training and test material. The attachment classification is assigned using a "majority vote" procedure where, for each child, 500 non-overlapping video frames contribute to the decision and the child is assigned to the attachment classification most frequently extracted from these frames.

The automatic classification was then compared to the classification from the detailed manual SAM rating conducted by trained MCAST raters.

## Results

Preliminary analyses with a series of chi-square tests showed that gender, school year (age) or school attended did not differ significantly for attachment security or attachment organization for either of SAM or MCAST.

Research Questions:

1. Is inter-rater reliability similar for both the MCAST and SAM?

*Inter-rater realibility for SAM*: the finalised ratings across the twice-rated SAM cases showed an agreement rate of 70% (Cohen's kappa .434) at the level of four-way attachment classifications; 77.6% (Cohen's kappa .49) for secure-insecure classifications and 95% (Cohen's *kappa* .64) for organised-disorganised classifications (see S1 Appendix).

*Inter-rater reliability for MCAST*: Of the 12 cases (20%) double-rated MCASTs, there were 3 discrepancies, therefore the MCAST data showed similar four-way inter-rater reliability (~75%) compared to SAM ratings at the four-way classification level.

Research Question 2:

2. Is there good agreement on two-way and four-way attachment classifications between SAM and the MCAST doll-play original when manually rated by trained raters?

As can be seen in Table 2, agreement between SAM and MCAST was excellent: secure versus insecure Cohen's *kappa* .73 (89% agreement, discordance n = 7); organised versus

**Table 2. Inter-measure agreement across MCAST and SAM (N = 61).**

| SAM ABCD Classifications | MCAST ABCD classifications | | | |
|---|---|---|---|---|
| | B-Secure | A-Insecure Avoidant | C-Insecure Ambivalent | D-Insecure Disorganised |
| | B-Secure | | | |
| B-Secure | **39** | 2 | 1 | 0 |
| A Insecure Avoidant | 0 | **2** | 1 | 0 |
| C Insecure Ambivalent | 2 | 0 | **8** | 0 |
| D Insecure Disorganised | 2 | 0 | 0 | **4** |
| Total (N = 61) | 43 | 4 | 10 | 4 |

*Cells which ratings are in agreement, are signified in bold.

**Table 3. Reports the classification performance in terms of accuracy (α), precision (π), recall (ρ) and F 1 measure.**

| α | π | ρ | F 1 |
|---|---|---|---|
| 80.2% | 67.4% | 91.2% | 77.5% |

disorganised Cohen's *kappa* .78 (97% agreement, discordance n = 2) and four-way agreement Cohen's *kappa* .73 (86% agreement, discordance n = 8). For cases on which SAM:MCAST ratings agreed (in bold in Table 2 below), 39 (73%) were secure, 2 (4%) insecure avoidant, 8 (15%) insecure ambivalent, and 4 (8%) insecure disorganised.

Research Question 3:

3. For SAM, is there good agreement between a machine learning algorithm and the manual classification of secure versus insecure attachment?

The overall agreement between the manual rating and the automatic rating for secure versus insecure attachment was 82.8%, i.e. for 87 of the 105 children, the same attachment classification was made with the automatic rating system and the manual rating system (see confusion matrix Table 3). Children with secure patterns were identified with greater accuracy than insecure children.

## Discussion

This study has shown that, using modern sensors and machine learning technology, it has been possible to develop the School Attachment Monitor (SAM) with: simple administration and portability compared to the MCAST doll-play original; manual rating that is similarly reliable to the MCAST doll-play original; an attachment distribution, using manual rating, that has good concordance with the MCAST and gives a similar distribution of four-way attachment classifications as the international literature [37] and, most importantly, a machine learning algorithm that gives an accurate classification of attachment security versus insecurity compared to manual ratings by trained MCAST raters.

The aim of our research programme, from its inception over a decade ago, was to develop a quick and easy measure of attachment that can be used in largescale public health monitoring or epidemiology. SAM has the potential to achieve that aim, although further research will be needed to examine its performance in a range of populations. Its administration is simple and the equipment required is cheap and portable. We would envisage that, for data collection purposes, schools in which there are consenting participants would be given a SAM kit for the day and the minimal research support required to ensure appropriate set up of the equipment. Throughout the day, participating children would take their turn to use SAM. From the child's perspective, this is usually an enjoyable task that takes around 20 minutes. Data could be collected in this way from several school classrooms in a week–or from even more classrooms if simultaneous data collection using more than one SAM set-up were conducted. This could allow largescale data collection (e.g. for tens of thousands of children) to take place in a relatively short space of time and with minimal resource.

We would envisage SAM being used for largescale mapping of attachment patterns in groups of children, not for individual attachment classifications at least at this stage. If individual attachment measurement is required–for example as part of a clinical assessment–then we would recommend that SAM might be used as a first-stage preliminary or screening assessment, followed by a more detailed assessment using a face to face measure such as the doll-play MCAST which would take into account the full spectrum and nuances of human interation, including narrative coherence [10]. Even then, a recent systematic review of attachment measures has recommended that assessment of a child's attachment pattern should never be

done (especially for court reports) using a single measure, even if well-validated measure, since no existing attachment measure has perfect psychometric properties when used alone [8]. Similarly, we would argue that even the well established attachment measures are not appropriate as single measures, for population screening. Insecure attachment classifications are not, in and of themselves, abnormal and are regarded by many as positive adaptations to less than optimal environmental circumstances [38]. Insecure attachment is, however, associated with increased risk of psychopathology of various types. There might therefore be potential for SAM, in future, to form part of an early school-based screening programme to identify children at risk of psychopathology, alongside other simple screening instruments, such as the Strengths and Difficulties Questionnaire [39]. If this were to be realized, such a screening programme would require careful development and testing.

Our study has certain limitations: despite achieving a good range of age gender and school deprivation category, the low recruitment rate means our study sample is unlikely to be representative of the general population, so the distribution of attachment classifications should be viewed with caution.

Artificial Intelligence approaches like those used in Roffo e al., 2019 [34] rely on statistical methodologies capable of associating variance in the measurable aspects of a phenomenon and variance in human judgments about the same phenomenon. In the case of attachment, this means that the methodologies associate variance in observable aspects of child behaviour and variance in the assessments provided by child mental health clinicians and/or attachment experts. In such a context, the main limitation is that the methodologies can work effectively only when the judgment of the experts is sufficiently consistent with the measurements available about the phenomenon. Therefore, whenever the measurements concern aspects that human observers do not take into account, at least indirectly, there is no guarantee that the approach can work. In addition, whenever the human observers make systematic mistakes (e.g., due to biases and/or limited skills), the approaches will tend to reproduce these, especially when they are consistent with the measurable aspects of behaviour. In addition to these limitations, AI approaches can work only as long as the experimental conditions remain similar (e.g., the sensors do not change and they are always in the same position, the characteristics of the children tend to remain stable, etc.). In other words, any machine learning algorithm is only as good as the data upon which it is based.

In future studies, there may be ways of refining algorithms. For example, using aspects of speech (e.g. amount or pitch), in addition to movement, might enhance the ability of future algorithms to discriminate between attachment classifications.

In order to realise the full potential of the SAM technology and to enable large-scale attachment studies, further technological development will be required—most critically, development of an automatic rating system which affords live rating of sessions for instant feedback. It will be important to explore the role of attachment to multiple caregivers (mother, father, and others) and the interplay between attachment patterns and other developmental (including genetic) factors. It is also critical that further testing should articulate the remit and role of SAM beyond typical development, such as in the context of adverse childhood experiences (ACEs) or developmental conditions, including neurodevelopmental disorders. The clinical utility of SAM would be enhanced if machine learning algorithms are able to discriminate between organised and disorganise attachment, but larger sample-sizes will be necessary for this to be achieved.

## Conclusions

We have developed a new tool with the potential to measure attachment at the population level, which has good reliability, produces a similar distribution of attachment classifications

compared to a well-validated attachment measure and has the potential for automatic rating.

## Supporting information

**S1 Fig. Computerised MCAST.** Parental consent was obtained for the use of the image. (DOCX)

**S1 Appendix.** (DOCX)

## Acknowledgments

With thanks to children and schools for participating and working with the research team.

## Author Contributions

**Conceptualization:** Alessandro Vinciarelli, Stephen A. Brewster, Helen Minnis.

**Data curation:** Maki Rooksby.

**Formal analysis:** Maki Rooksby, Simona Di Folco, Mohammad Tayarani, Dong-Bach Vo, Alessandro Vinciarelli, Stephen A. Brewster.

**Funding acquisition:** Alessandro Vinciarelli, Stephen A. Brewster, Helen Minnis.

**Investigation:** Maki Rooksby, Dong-Bach Vo, Rui Huan, Alessandro Vinciarelli, Stephen A. Brewster, Helen Minnis.

**Methodology:** Maki Rooksby, Simona Di Folco, Mohammad Tayarani, Dong-Bach Vo, Alessandro Vinciarelli, Stephen A. Brewster, Helen Minnis.

**Project administration:** Maki Rooksby.

**Supervision:** Simona Di Folco, Alessandro Vinciarelli, Stephen A. Brewster, Helen Minnis.

**Validation:** Rui Huan.

**Writing – original draft:** Maki Rooksby, Helen Minnis.

**Writing – review & editing:** Maki Rooksby, Simona Di Folco, Mohammad Tayarani, Dong-Bach Vo, Rui Huan, Alessandro Vinciarelli, Stephen A. Brewster, Helen Minnis.

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
