## [Decision Letter · Decision Letter 0]

31 Dec 2020

PONE-D-20-29493

The School Attachment Monitor - a novel computational tool for assessment of attachment in middle childhood.

PLOS ONE

Dear Dr. Minnis,

Thank you for submitting your manuscript to PLOS ONE. After careful consideration, we feel that it has merit but does not fully meet PLOS ONE’s publication criteria as it currently stands. Therefore, we invite you to submit a revised version of the manuscript that addresses the points raised during the review process.

Both reviewers agree that the language style needs to be revised thoroughly and even more important both raise methological questions that need to be addressed in a revision that I look forward to read. Please be aware that machine learning methods are still new to the field and need to be explained in more details than more common methods.

We look forward to receiving your revised manuscript.

Kind regards,

Svenja Taubner

Academic Editor

PLOS ONE

Journal Requirements:

3. Please amend the manuscript submission data (via Edit Submission) to include author Rui Huan.

4. We note that Figure 4 and Supplementary figure include an image of a participant in the study.

Reviewers' comments:

Reviewer's Responses to Questions

**Comments to the Author**

1. Is the manuscript technically sound, and do the data support the conclusions?

Reviewer #1: Partly

Reviewer #2: Partly

2. Has the statistical analysis been performed appropriately and rigorously? 

Reviewer #1: Yes

Reviewer #2: I Don't Know

3. Have the authors made all data underlying the findings in their manuscript fully available?

Reviewer #1: No

Reviewer #2: Yes

4. Is the manuscript presented in an intelligible fashion and written in standard English?

Reviewer #1: Yes

Reviewer #2: Yes

5. Review Comments to the Author

Reviewer #1: First of all, I applaud the authors for the incredible amount of effort that has been put into this project. While the author’s work could prove a valuable asset in making attachment research more feasible in large scale studies, there are nevertheless several important limitations that I feel need to be addressed.

First, the language style of the manuscript seems loose. Bullet points are used frequently and there are a lot of expressions that give the manuscript a feel of more ordinary speech rather than formal writing.

An example of this is the end of the first paragraph on page 15: …” childhood attachment research (20), probably due to the technical challenges of doing so.” Where the speculation at the end of the sentence are not further contextualized or substantiated and also don’t serve a particular purpose. In the same vein, the next paragraph continues with “Around the same time,…” followed with a direct quotation that could have been paraphrased instead. There are several more examples like this throughout the manuscript. If possible I would also integrate most of the bullet points into the text to make it more seamless.

In the methods section there are several core bits of information missing.

While the development of the procedure is for the most part well described, important details such as a confusion matrix of the both the manual ratings of SAM vs MCAST as well as the manual SAM ratings vs the algorithms ratings should be added. This would allow the reader to more accurately gauge where the strengths and weaknesses of the different rating systems lie, where they over estimate one attachment classification over the other and so on. Also the Machine Learning algorithm along with its tuning parameters is not described at all. While this might be because of issues with the overall length of the manuscript, in this case please provide an appendix with the appropriate information. From the text presented it is unclear which kind of algorithm was used (neural-nets? Random Forests? Gradient Boosting? If yes which variant?) as well as how the hyperparameters where set up and what the end values were for those hyperparameters. Also, there should be a sentence justifying choosing leave-one-out cross validation over k-fold cross-validation along with a sentence in the limitations section on how this method of cross-validation is probably overestimating the generalizability of the algorithm because of the small sample size and lack of a separate validation set.

I hope the authors find these comments helpful and not too discouraging as this an excellent work which just needs some polishing.

Reviewer #2: I am very sympathetic to the exciting and pioneering line of research presented in this manuscript. I feel strongly that it has the potential to "push the envelope" within the attachment research and allow these concepts to leave a greater empirical footprint than they currently do and more informatively enter common parlance for clinical assessments. In that sense, I share the authors' enthusiasm about their truly impressive research and validation efforts emerging from many years of arduous and no doubt painstaking development.

However, that said, based on the style of this manuscript, I would like to caution the authors about underestimating the difficulty of this task and the importance of subjecting their work to an exceedingly high standard of theoretical and empirical scrutiny, especially if they intend to advance the bold claim of having developed a "new attachment measure" including an algorithm-based rating system premised on machine-learning. If they do not do so, I predict a lukewarm acceptance of their work within the attachment community and this, of course, would be inimical to their objective. While I understand the aim to promote the new instrument, I had a number of concerns regarding (a.) the lack of theoretical and methodological rigour in evaluating the SAM, especially in the results and discussion and (b.) the language used throughout the manuscript which I feel needs to be considerably toned down against the backdrop of what I would still consider to still be preliminary findings. My hunch is that more cautious language and a theoretical section that offers some thoughts on the boundaries inherent in computerizing such an essentially interpersonal task as story-completion at such a young age would ultimately convince more readers. I will spell out below what exactly I mean by this in the order of importance.

1. From a theoretical perspective the discussion would benefit substantially from discussing two related issues at length:

First, as far as computerised administration using the SAM is concerned, what drawbacks to the authors expect in which populations? More specifically, to the extent that story-completion is inherently an interpersonal paradigm providing access to attachment representations (and builds on the cooperative principle and the associated Gricean maxims) is it actually realistic to assume that the same communication patterns will emerge when children tell their stories to a computer vs. an experimenter? In fact, is it conceivable that some attachment patterns (especially avoidant children) are blurred due to this less interpersonal mode of measuring attachment because the procedure is less anxiety provoking and/or creates fewer opportunities for co-regulation by the experimenter? Moreover, most story-completion procedures use a highly interactive warm-up phase as well as interactive prompts in the event that children avoid certain core issues or dilemmas in the story. Children's responsiveness to these interactive prompts are a key factor in administering the task (because the child get's the sense of a real play interaction) as well in coding the task. From this vantage point, is it very far-fetched to assume that the absence of these features may result in a very different measure compared to other story-completion measures? To be clear, I am not arguing that the SAM does not measure an interesting and important construct, but I would be hesitant based on a single study to make strong claims that we are tapping into parallel (though perhaps related) processes with SAM and the MCAST. To me, the high levels of concordance between the two measures are not sufficient to make such a claim, especially in light of the partial non-independence of the ratings (see below).

Second, as far as the machine learning algorithm is concerned, I found myself wondering whether any measure based on measurement of movement and distances can ever do more than provide a behavioural "correlate" or "marker" of attachment representations, as indexed by story-completions. The point is that the machine-learning algorithm captures an entirely different level of information and is "ignorant" of the meaning and representations the child conveys in the story. While it is fascinating that there may be some overlap between content and behavior, I would shy away from equating the two as the authors do when they refer to the machine-learning algorithm as an automated means of classifying or rating "attachment". Analogously, I would venture that no neuroscientist would equate certain patterns of neural activity with an attachment classification, but would rather and more conservatively refer to physiological outcomes, correlates or markers (depending on the state of evidence, see Cacioppo et al., 2017, Handbook of Psychophysiology). I would therefore strongly urge the authors to adopt a similar more conservative terminology in describing the results of their machine learning algorithm, especially as it remains unclear if the results generalise to other (especially at-risk) samples and also show overlap with other attachment measures that have received more validation and cover a similar age-range (e.g., CAI; see Jewell et al., 2019).

2. With regard to the methods and results, I felt many important questions were left unanswered:

a.) on p. 12 the authors state that to their surprise children were "perfectly capable of a) understanding and using a two dimensional plan version of the dolls-house".This is obviously a casual observation, but what exactly does "perfectly capable" mean? Did the order of presentation (fuzzy felt map first group vs. MCAST first group) influence the quality of narration or not? After all, it is conceivable that those who first received the MCAST had warmed up more effectively and therefore were better able to "step it up a notch" and also narrate stories in the two-dimensional space of the fuzzy felt map. The same question applies later on again when the authors counterbalanced SAM and MCAST in their n=61 sample for Phase 2. Did order influence their results in any way in either of these samples?

b.) On p. 14 the authors report this result: "The overall attachment classification for SAM did not differ depending on which version of the prototype was administered: (Spearman’s rho (116) = 0.065, N.S.)." Maybe I missed something, but I thought attachment classification and prototype version were both categorical measures. Is Spearman's rho appropriate for categorical measures?

c.) Information on the sample is very sparse. Table 1 does not divide the information on age and gender-distribution up by Phase of the Study. Therefore, I could not glean whether any age-group or gender was under- or over-represented during any phase of the study.

d.) As far as I could tell, for the manual rating in some cases the expert rater was consulted for the SAM and MCAST on the same child? Hence, coding on these cases was not independent (even if the ratings were completed on different days) and I would like to see the results before the discordance was abolished with the help of the expert rater - after all, this is more likely to reflect the result that would be obtained outside of a validation study.

e.) Two children were excluded from the algorithm-based approach due insufficient data-quality. I think it is justified to argue that these children should be included when reporting the results because they would ultimately lead to ultimately discordance (similar to "cannot classify" in the AAI)

f.) The information provided on the algorithm is very sparse. Instead the authors cite a conference paper by Roffo and colleagues that cannot be accessed online. Specifically, it remained unclear to me what "use of the entire dataset" meant? Did the algorithm also have access to and use information other than the videos, i.e., did it "know" the attachment classification of the child and try to predict it from the information in the video?

Or did the algorithm "learn" based purely on the information drawn from the videos.

The authors must provide many more details in an Appendix or I will not be in a good position to make a call as to the significance of their findings.

g.) Was age, gender or deprivation index score related to the concordance between MCAST and SAM and between machine-learning algorithm and MCAST?

3. Comment on Language:

The manuscript is replete with comments, such as this:

"From the user perspective, these modifications simplified SAM set-up compared to CMCAST since story stems

could potentially be delivered on any school computer (laptop or PC) installed with our SAM software, and the only other specific “kit” required would be the dolls-house mat, furniture, computerised dolls and the webcam-like camera with depth sensor. These are light and highly portable compared to the MCAST original set-up."

As a whole I couldn't escape the feeling that this manuscript is too oriented to promoting the new measure and explaining the practilaties of this measure. As consequence, I got the sense of a user manual manual or even a commercial website without the necessary self-critical, dispassionate attitude necessary for a rigorous empirical examination. I think it would already help a great deal, as I mentioned above, if the authors delved more into the difficulties and risks inherent in transferring an interactive story-completion task to a computerized environment and then subjecting the video-based data to a computerized algorithm. Also, it would importnatn to acknowledge that the most authoritative review of attachment measures in the field actually states that middle childhood and adolescence are phases for which "no gold-standard measure exist" (p. 72). Hence, calling the MCAST a gold-standard does not seem appropriate and comparing the results of the SAM with other more dimensional story-completion procedures, such as the ASCT or the MSSB would make sense.

4. Finally, regarding clinical utility of the SAM, I have reservations about believing that the SAM would indeed yield comparable levels of storytelling to MCAST (or other story-stem techniques) in non-typically developing children. My primary concerns, some of which I am reiterating here, would be that (a.) the cognitive demands placed on the child in the SAM presumably exceed those required for the MCAST (b.) the task cannot be tailored to the child's individual skill level (c.) no warm-up procedure could be implemented, and (d.) to the extent that story-telling reflects an interpersonal process between the child and the experimenter the SAM presumably only captures this aspect insufficiently (as mentioned above, but potentially even more relevant in clinical populations). As for the machine-learning based algorithm, its clinical utility will be enhanced substantially if it proves sensitive to identifying behavioural correlates of disorganization.

6. PLOS authors have the option to publish the peer review history of their article (what does this mean?). If published, this will include your full peer review and any attached files.

Reviewer #1: No

Reviewer #2: No

---

## [Author Response · Author response to Decision Letter 0]

25 Feb 2021

Dear Professor Taubner,

PONE-D-20-29493

The School Attachment Monitor - a novel computational tool for assessment of attachment in middle childhood.

Thank you for inviting us to resubmit this paper, for the helpful comments of the reviewers and for your own helpful overview of the main points we should address. For clarity, we have included each of the points made in italics and have now relabelled our files according to PLOS ONE guidelines.

You had a specific question about the availability of our data. Unfortunately, there are ethical restrictions to sharing our data publicly: although we recognise that data sharing would support open science and reproducibility, parental consent for data sharing was not sought since our dataset comprises identifiable video data of children. Even the secondary data (for the manual ratings of MCAST and SAM and for the machine learning) are derived directly from video data and any secondary analysis would only be reproducible with reference to the original videos, so we cannot share a de-identified secondary dataset.

3. Please amend the manuscript submission data (via Edit Submission) to include author Rui Huan.

Thank you for drawing our attention to this oversight which we have now corrected. 

4. We note that Figure 4 and Supplementary figure include an image of a participant in the study.

As reported in the manuscript, we obtained consent from all parents and guardians of the children who participated in the study. As a separate optional consent, parents were asked whether we may share children’s video recordings/stills for academic purposes such as in publications of study results, presentation at academic events such as conferences or teaching university students. The blurred image of a child which appears in our manuscript is of a child whose parents has given this additional optional consent. Even having obtained this parental consent, we have done our best to prevent identification of the child shown. This image is already in the public domain, having already appeared in our earlier publication specifically relating to the technical details of the development of the machine learning algorithms (i.e., Roffo et al., 2019). We have now included in the Figure legend that parental consent was obtained for the use of the image and hope that is acceptable.

This has now been done.

Reviewer #1: First of all, I applaud the authors for the incredible amount of effort that has been put into this project. 

Thank you for these kind words.

While the author’s work could prove a valuable asset in making attachment research more feasible in large scale studies, there are nevertheless several important limitations that I feel need to be addressed. First, the language style of the manuscript seems loose. Bullet points are used frequently and there are a lot of expressions that give the manuscript a feel of more ordinary speech rather than formal writing.

We have reviewed the entire document and amended the language in many places, removing any speculation and doing our best to be clear and factual throughout.

An example of this is the end of the first paragraph on page 15: …” childhood attachment research (20), probably due to the technical challenges of doing so.” Where the speculation at the end of the sentence are not further contextualized or substantiated and also don’t serve a particular purpose.

This statement has now been removed as we agree that it did not add anything.

In the same vein, the next paragraph continues with “Around the same time,…” followed with a direct quotation that could have been paraphrased instead. There are several more examples like this throughout the manuscript. 

This has now been paraphrased in a more formal style.

If possible I would also integrate most of the bullet points into the text to make it more seamless.

This has been changed.

In the methods section there are several core bits of information missing.

While the development of the procedure is for the most part well described, important details such as a confusion matrix of the both the manual ratings of SAM vs MCAST 

This is in Table 2.

…as well as the manual SAM ratings vs the algorithms ratings should be added. 

This has now been added to the manuscript (Table 3).

This would allow the reader to more accurately gauge where the strengths and weaknesses of the different rating systems lie, where they over estimate one attachment classification over the other and so on. 

This is an interesting point. We stated in both this paper and our previous publication (Roffo et al) that the machine learning algorithm is more precise for secure compared to insecure attachment, but we have not discussed this in terms of four-way classifications because further work with larger numbers would be required to confidently state whether SAM over- or under-estimates any particular attachment classification. This would certainly be interesting to explore in future studies.

Also the Machine Learning algorithm along with its tuning parameters is not described at all. While this might be because of issues with the overall length of the manuscript, in this case please provide an appendix with the appropriate information. From the text presented it is unclear which kind of algorithm was used (neural-nets? Random Forests? Gradient Boosting? If yes which variant?) as well as how the hyperparameters where set up and what the end values were for those hyperparameters. Also, there should be a sentence justifying choosing leave-one-out cross validation over k-fold cross-validation along with a sentence in the limitations section on how this method of cross-validation is probably overestimating the generalizability of the algorithm because of the small sample size and lack of a separate validation set.

We anticipate that this paper will be most of interest to psychologists and psychiatrists so decided not to go into too much detail about the technicalities of our machine learning algorithm development in the text, especially because we have a separate publication (Roffo et al) that is referenced and describes this in detail. What we have now done is add a concise technical appendix because we agree that readers should not have to seek another publication to find these details.

Reviewer #2: I am very sympathetic to the exciting and pioneering line of research presented in this manuscript. I feel strongly that it has the potential to "push the envelope" within the attachment research and allow these concepts to leave a greater empirical footprint than they currently do and more informatively enter common parlance for clinical assessments. In that sense, I share the authors' enthusiasm about their truly impressive research and validation efforts emerging from many years of arduous and no doubt painstaking development.

Thank you for these encouraging words.

I would like to caution the authors about underestimating the difficulty of this task and the importance of subjecting their work to an exceedingly high standard of theoretical and empirical scrutiny, especially if they intend to advance the bold claim of having developed a "new attachment measure" including an algorithm-based rating system premised on machine-learning. If they do not do so, I predict a lukewarm acceptance of their work within the attachment community and this, of course, would be inimical to their objective. This very helpful comment has caused us to return to the early attachment literature, especially the work of Bowlby and Schaffer, and we believe this has allowed us to write introduction and discussion sections that are now on a more sound theoretical footing. Specifically, we have returned to Bowlby’s 1958 work in which he acknowledges that the human ability for symbolism could be key to attachment but that this is unlikely, and that the instinctual motor function of proximity-seeking seen across species is more likely to be of central importance. We have tried to be cautious, throughout, about the nature of our empirical findings and to point the reader wherever possible to areas that will require further exploration.

1. From a theoretical perspective the discussion would benefit substantially from discussing two related issues at length:

First, as far as computerised administration using the SAM is concerned, what drawbacks to the authors expect in which populations? More specifically, to the extent that story-completion is inherently an interpersonal paradigm providing access to attachment representations (and builds on the cooperative principle and the associated Gricean maxims) is it actually realistic to assume that the same communication patterns will emerge when children tell their stories to a computer vs. an experimenter?

Thank you for highlighting this. We have elaborated on these points in the Discussion as suggested and think the paper is improved as a result. When developing the CMCAST, we also expected that it might not be realistic for children to tell their story to a computer with sufficient richness to allow attachment classification. However, it was quickly evident that key attachment-related indices – specifically arousal, proximity-seeking and assuagement - were central to the rating system rather than more subtle aspects of the narrative. So although we agree that the MCAST – and SAM – are interpersonal measures, we have been constantly struck by the apparently instinctual nature of the children’s story-completion. This is in contrast to story-stem completion by children who are older than the recommended MCAST age-range, who seem to be more performative in their story-telling. This observation, and the original writings of Bowlby, have caused us to suspect that the story content may not be relevant in attachment classification. This would chime with the findings from Adult Attachment Interview rating which, as the reviewer says, is based on form rather than content of the narrative. It is possible that the contraventions of Grice’s maxims, such as hesitancy, observed in disorganised AAI narratives could be fundamentally a motor phenomenon, but clearly much more research would be required to establish if this is the case.

Lastly, according to Reeves and Nass, the brain is unable to tell the difference between humans and artificial agents represented on computer and this is the basis of the explosion of avatar-use in computer-games and social media sites. This is described in their book “The Media Equation” which we have now referenced in the text.

In fact, is it conceivable that some attachment patterns (especially avoidant children) are blurred due to this less interpersonal mode of measuring attachment because the procedure is less anxiety provoking and/or creates fewer opportunities for co-regulation by the experimenter?

This is a very interesting question, but we have not gone into individual attachment classifications in any detail in the paper because we would require larger numbers to explore such questions meaningfully. Again, this would be an interesting avenue to explore in future research.

Moreover, most story-completion procedures use a highly interactive warm-up phase as well as interactive prompts in the event that children avoid certain core issues or dilemmas in the story. Children's responsiveness to these interactive prompts are a key factor in administering the task (because the child gets the sense of a real play interaction) as well in coding the task. From this vantage point, is it very far-fetched to assume that the absence of these features may result in a very different measure compared to other story-completion measures?

Thank you for highlighting this important aspect and we have now included further details of this in the text of the paper, rather than simply referring the reader to a previous publication. In order to orientate the child user towards interacting with the computer in SAM, a cat character/avatar greets the child, asks the child to position themselves so that they are captured well by the camera of the SAM set-up via a game of “posing for a photo”, as well as explaining to the child what their task is, and introducing the Breakfast vignette, just as in MCAST. SAM and MCAST both have the same vignettes and, in both, the Breakfast vignette is a warm-up task that does not contribute to attachment classification. Administrators of MCAST are discouraged from indulging in additional warm-up tasks, since poor engagement at this stage is information that has to be taken into account when coding. We have now inserted further clarification on this in the Methods section.

To be clear, I am not arguing that the SAM does not measure an interesting and important construct, but I would be hesitant based on a single study to make strong claims that we are tapping into parallel (though perhaps related) processes with SAM and the MCAST. To me, the high levels of concordance between the two measures are not sufficient to make such a claim, especially in light of the partial non-independence of the ratings (see below).

This is the second study in which we have found excellent concordance with computerised delivery of MCAST story-stems and an almost identical distribution of attachment patterns to the MCAST classifications of the same group of children (see also Minnis et al 2010 https://doi.org/10.1002/mpr.324). We think we have now made a much stronger theoretical argument for SAM as an attachment measure. Despite that, we have been cautious in our claims about the psychometric properties of SAM and, throughout, have stressed that further research would be needed before we could recommend it for use in either clinical work or screening.

Second, as far as the machine learning algorithm is concerned, I found myself wondering whether any measure based on measurement of movement and distances can ever do more than provide a behavioural "correlate" or "marker" of attachment representations, as indexed by story-completions. The point is that the machine-learning algorithm captures an entirely different level of information and is "ignorant" of the meaning and representations the child conveys in the story. 

Bowlby proposed that the fundamental feature of attachment, in humans and other animals, is movement and that features of children’s symbolic play might or might not be of additional importance in humans (Bowlby, 1958). Even if features of the narrative turn out to be useful for manual rating, that does not necessarily mean that a “marker” based largely on movement would not be a useful measure. The usefulness of markers, even when ignorant of meaning, has been described in the field of Social Signal Processing (Vinciarelli et al, 2009). Markers are often regarded as the holy grail of medical measurement, especially if such a marker gives just enough information in a simple and easy way without necessarily requiring burdensome assessment. An example would be blood pressure: extremely high blood pressure, i.e. “malignant hypertension”, is a medical emergency that could be caused by a range of underlying pathologies. Similarly, there are thought to be various underlying causes of attachment disorganisation (e.g. bereavement, abuse, parental unresolved trauma) but a reliable marker of disorganisation would still be useful even if unable to explain what underpinned the disorganisation in any individual.

While it is fascinating that there may be some overlap between content and behavior, I would shy away from equating the two as the authors do when they refer to the machine-learning algorithm as an automated means of classifying or rating "attachment". I would therefore strongly urge the authors to adopt a similar more conservative terminology in describing the results of their machine learning algorithm, especially as it remains unclear if the results generalise to other (especially at-risk) samples and also show overlap with other attachment measures that have received more validation and cover a similar age-range (e.g., CAI; see Jewell et al., 2019).

The fact that SAM, just like in MCAST, presents a child with a stressful story-stem then sets the child the task of finishing the story (i.e. resolving an attachment-related dilemma) means that the SAM “front-end” is clearly presenting the child with an attachment stimulus. Based on our reading of the early attachment literature, we do not think it would be reasonable for us to describe the “back-end” machine-learning classification as anything other than an attachment classification simply because it is based on the movement of the dolls. Regarding other populations, we did extensive cross-validation with MCAST in our previous study (Minnis 2010) and we have been clear, in the discussion of this current paper, about the need for further research in at-risk and clinical samples.

2. With regard to the methods and results, I felt many important questions were left unanswered:

a.) on p. 12 the authors state that to their surprise children were "perfectly capable of a) understanding and using a two dimensional plan version of the dolls-house".This is obviously a casual observation, but what exactly does "perfectly capable" mean? Did the order of presentation (fuzzy felt map first group vs. MCAST first group) influence the quality of narration or not? After all, it is conceivable that those who first received the MCAST had warmed up more effectively and therefore were better able to "step it up a notch" and also narrate stories in the two-dimensional space of the fuzzy felt map. The same question applies later on again when the authors counterbalanced SAM and MCAST in their n=61 sample for Phase 2. Did order influence their results in any way in either of these samples?

As reported, the presentation order of the two versions were counterbalanced across the participants. We did not find that any child in Workshop 1 was unable to respond to the story-stem task presented to them, regardless which version they experienced first. As the reviewer points out, it is indeed possible that children may have felt more familiar with the second version that they received. However, we did not find any order effects in the quality of their narrative or engagement with the task and we have clarified the language so that this is hopefully now clearer in the paper.

b.) On p. 14 the authors report this result: "The overall attachment classification for SAM did not differ depending on which version of the prototype was administered: (Spearman’s rho (116) = 0.065, N.S.)." Maybe I missed something, but I thought attachment classification and prototype version were both categorical measures.

Thank you for noticing this error. Your comment caused us to reflect that, since only very minor modifications were made to SAM, it is meaningless to include statistical testing here and we have removed this sentence.

c.) Information on the sample is very sparse. Table 1 does not divide the information on age and gender-distribution up by Phase of the Study. Therefore, I could not glean whether any age-group or gender was under- or over-represented during any phase of the study.

We have now made it clearer that this study did not aim to be population-representative but, rather, to achieve sampling across a wide range of socio-economic status and across the full age-range targeted by the MCAST. We have included further detail in the discussion to remind the reader that we did not set out to evaluate the distribution of attachment classifications in this population and that this would be an important task for future research.

d.) As far as I could tell, for the manual rating in some cases the expert rater was consulted for the SAM and MCAST on the same child? Hence, coding on these cases was not independent (even if the ratings were completed on different days) and I would like to see the results before the discordance was abolished with the help of the expert rater - after all, this is more likely to reflect the result that would be obtained outside of a validation study.

You are correct that we went through a very rigorous manual rating process in which all manual ratings in phase 1 were checked by an expert rater. As we have explained in the text, this was partly because, in the early phases of the study, our raters were fairly inexperienced and we realised that high quality manual rating was essential if we were to then test our manual ratings against the machine learning algorithm. Although our expert raters (SDF and HM) were occasionally consulted for the SAM and MCAST on the same child, the manual rating was conducted over several months and the project manager (MR) ensured that videos on the same child were never offered to an expert raters within the same rating “batch”. Since both expert raters were conducting ratings in evenings and weekends over and above other full time jobs, they could not have remembered details of videos they rated in previous batches so we are confident that the ratings were independent. We had already, in our previous publication on CMCAST (Minnis 2010), demonstrated that trained MCAST raters were just as reliable in rating the MCAST and CMCAST so our main task, in this current study, was to ensure that we achieved the highest quality SAM ratings to then test against the machine learning algorithm.

e.) Two children were excluded from the algorithm-based approach due insufficient data-quality. I think it is justified to argue that these children should be included when reporting the results because they would ultimately lead to ultimately discordance (similar to "cannot classify" in the AAI)

We beg to differ regarding this since, in face-to-face story-stem studies, it would be usual to exclude children as unrateable who did not engage with the task as stated in the MCAST manual. This differs from “cannot classify” in AAI coding since, in this latter case, the adult participant would have engaged with the task, yet their transcript is still unclassifiable. It is rare for AAI testing to result in complete non-engagement but, by coincidence, one of the authors has reported this previously (Minnis, 1996, The Journal of Nervous and Mental Disease: July 1996 - Volume 184 - Issue 7 - p 440).

f.) The information provided on the algorithm is very sparse. Instead the authors cite a conference paper by Roffo and colleagues that cannot be accessed online. Specifically, it remained unclear to me what "use of the entire dataset" meant? Did the algorithm also have access to and use information other than the videos, i.e., did it "know" the attachment classification of the child and try to predict it from the information in the video?

Or did the algorithm "learn" based purely on the information drawn from the videos.

The authors must provide many more details in an Appendix or I will not be in a good position to make a call as to the significance of their findings.

This is a very good point and we have now addressed this by including a technical summary.

g.) Was age, gender or deprivation index score related to the concordance between MCAST and SAM and between machine-learning algorithm and MCAST?

There were no apparent differences in MCAST/SAM concordance or between machine-learning algorithm/SAM concordance according to these demographic variables, but since this level of sub-grouping of our sample resulted in small numbers within cells we thought it would have been inappropriate to focus on this in the paper or to conduct statistical testing which would inevitably have given null results that could have been misleading. So although we do not have any indication from our findings that concordance is likely to be influenced by age, gender or deprivation index, it would certainly be wise for this to be explored in more detail in future research.

3. Comment on Language:

The manuscript is replete with comments, such as this:

"From the user perspective, these modifications simplified SAM set-up compared to CMCAST since story stems could potentially be delivered on any school computer (laptop or PC) installed with our SAM software, and the only other specific “kit” required would be the dolls-house mat, furniture, computerised dolls and the webcam-like camera with depth sensor. These are light and highly portable compared to the MCAST original set-up."

As we described in the paper, we sought the user perspective in workshops and during data collection so these statements are factual and would be helpful for any future researcher deciding on the feasibility of using SAM in future studies.

As a whole I couldn't escape the feeling that this manuscript is too oriented to promoting the new measure and explaining the practicalities of this measure. As consequence, I got the sense of a user manual or even a commercial website without the necessary self-critical, dispassionate attitude necessary for a rigorous empirical examination. 

We have been very careful to be dispassionate and empirical and believe these helpful reviewers’ comments have allowed us to strengthen theoretical underpinning and methodological clarity of the paper. 

I think it would already help a great deal, as I mentioned above, if the authors delved more into the difficulties and risks inherent in transferring an interactive story-completion task to a computerized environment and then subjecting the video-based data to a computerized algorithm.

This is a difficult balance to achieve: for example it is for this very reason that we have gone into our previous work with CMCAST in some detail in the introduction so that the reader can see that this paper is simply a step in a long programme of work. However, it is challenging to report all the pitfalls and risks without losing the focus of the current paper. We hope that, with the help of the comments from both reviewers, we have now struck this balance.

Also, it would important to acknowledge that the most authoritative review of attachment measures in the field actually states that middle childhood and adolescence are phases for which "no gold-standard measure exist" (p. 72).

We agree. We have now removed all mentions of the term “gold standard” and have replaced it with terms such as “well-regarded and highly cited”.

Hence, calling the MCAST a gold-standard does not seem appropriate and comparing the results of the SAM with other more dimensional story-completion procedures, such as the ASCT or the MSSB would make sense.

Other studies have already shown that the MCAST can also be regarded as a continuous measure (please, see Di Folco, S., Messina, S., Zavattini, G.C. et al. Attachment to Mother and Father at Transition to Middle Childhood. J Child Fam Stud 26, 721–733 (2017). https://doi.org/10.1007/s10826-016-0602-7), assessing the child’s secure base script knowledge.

4. Finally, regarding clinical utility of the SAM, I have reservations about believing that the SAM would indeed yield comparable levels of storytelling to MCAST (or other story-stem techniques) in non-typically developing children.

The rating systems of MCAST and other story-stem techniques such as MSSB do not focus on the skill of delivering a story as such but, rather, on representation of attachment-related phenomena such as assuagement of distress and coherence of narrative. These measures are recommended for use with children with a developmental age of four years and upwards, and we have found in our previous studies of computerised story stem measures (CMSSB, Minnis et al 2006 and CMCAST, Minnis et al 2010) that the computerised versions are appropriate for the same developmental age and result in ratings that are just as reliable.

My primary concerns, some of which I am reiterating here, would be that (a.) the cognitive demands placed on the child in the SAM presumably exceed those required for the MCAST.

Humans interact with computers in very similar ways to the way they interact with humans, as described in the seminal book “The Media Equation: How People Treat Computers, Television, and New Media Like Real People and Places”, Reeves and Nass, 2003. Interacting “socially” with a computer does not, therefore, appear to entail greater cognitive demands. Both our current and previous work (Minnis et al, 2010) showing that the computerised version of the MCAST is as reliable as the dolls-house version would support that. 

(b.) the task cannot be tailored to the child's individual skill level 

Administrators of non-computerised story-stem tasks would be discouraged from tailoring their administration to the child’s individual skill level. If the child was unable to understand the standard prompts, s/he would be regarded as not being at the appropriate developmental level for the task. Tailoring the task would likely reduce the reliability of the task, therefore being unable to do so in a computerised version is a potential an advantage in terms of replicability, as we have described previously (Minnis, 2006 and 2010).

(c.) no warm-up procedure could be implemented, and

See above

(d.) to the extent that story-telling reflects an interpersonal process between the child and the experimenter the SAM presumably only captures this aspect insufficiently (as mentioned above, but potentially even more relevant in clinical populations).

Interpersonal processes are undoubtedly essential in ensuring that a child is relaxed and ready to take part in a research assessment, but such processes are just as important whether preparing a child to be assessed using a paper, play-based or computerised measure. Story-stem measures are not intended to engage interpersonal processes between the child and the experimenter – neither the MCAST nor SAM is designed to capture relational processes with the experimenter but between the child doll and mummy doll. Differences in the interpersonal styles of administrators are a potential source of poor replicability which (as we have commented previously in Minnis 2006 and 2010) might be a benefit of the computerised versions. 

As for the machine-learning based algorithm, its clinical utility will be enhanced substantially if it proves sensitive to identifying behavioural correlates of disorganization.

This is an important point and we have now highlighted this at the end of the discussion.

Thanks again for these very helpful comments. We hope you agree that addressing these has improved the paper,

Yours sincerely,

Helen Minnis

---

## [Decision Letter · Decision Letter 1]

18 May 2021

PONE-D-20-29493R1

The School Attachment Monitor - a novel computational tool for assessment of attachment in middle childhood.

PLOS ONE

Dear Dr. Minnis,

Thank you for submitting your manuscript to PLOS ONE. After careful consideration, we feel that it has merit but does not fully meet PLOS ONE’s publication criteria as it currently stands. Therefore, we invite you to submit a revised version of the manuscript that addresses the points raised during the review process.

The reviewer is an expert in story stem assessments and made excellent points to revise the manuscript. I agree with the points that the limitations of a solely behavioral assessment need to be addressed more thoroughly and that readers should get more detailed information on the algorythms behind your analysis. I feel confident that the manuscripts will be a valid contribution to the PLOS One readership after this revision.

We look forward to receiving your revised manuscript.

Kind regards,

Svenja Taubner

Academic Editor

PLOS ONE

Journal Requirements:

Reviewers' comments:

Reviewer's Responses to Questions

**Comments to the Author**

1. If the authors have adequately addressed your comments raised in a previous round of review and you feel that this manuscript is now acceptable for publication, you may indicate that here to bypass the “Comments to the Author” section, enter your conflict of interest statement in the “Confidential to Editor” section, and submit your "Accept" recommendation.

Reviewer #2: (No Response)

2. Is the manuscript technically sound, and do the data support the conclusions?

Reviewer #2: Yes

3. Has the statistical analysis been performed appropriately and rigorously? 

Reviewer #2: Yes

4. Have the authors made all data underlying the findings in their manuscript fully available?

Reviewer #2: No

5. Is the manuscript presented in an intelligible fashion and written in standard English?

Reviewer #2: Yes

6. Review Comments to the Author

Reviewer #2: I appreciate the authors thorough and interesting response letter. In my view, although many of their points are well taken, too little of the pros and cons discussed in the response letter have made their way into the revised manuscript. In fact, I firmly believe that the manuscript will make a more lasting impact if the authors show readers that they are aware of these issues, by unpacking the concerns that may be raised by their task and automated ratings and pitting their own counter-arguments against these possible concerns. I do not think glossing over these issues will do them any favours. Essentially, I am thus advocating for significantly extending the discussion and limitations sections by addressing the following issues:

1. Ever since narrative measures have become popular in attachment research in the mid-1980s with Mary Main's move to the level of representation, they have been considered windows to children's internal representations. In so doing, they involve tapping into children's processes of meaning-making, by which I mean they are thought to provide access to the child's subjective perspective on interactions and relationships with their caregivers and themselves. As such, narrative measures put us in a much better position to understand what Bowlby conceived of as the "goal-corrected partnership" that eventually develops between caregivers and children and involves a basic capacity for perspective-taking. From this perspective, a purely proximity-seeking and movement-based algorithm carries the inherent risk of implying that we can safely ignore this information and a sole focus on behavior (akin to the Strange Situation) will do. Do the authors really want to suggest this, i.e., that the same rating principles apply to story-completions as they do to the strange situation? Forgive me for saying so, but that would somehow feel a bit like putting the narrator in a Skinner box. From their response letter, however, I gathered that they are not advancing such a bold claim, but, rather, alerting readers to the possibility that the assessment of movement and behavioural manipulation of the figures may offer valuable insights with attachment-relevant implications.

2. While the authors nicely illustrate the warm-up procedure used in the SAM, the lack of standardized prompting procedures (which by the way is a standardized way of tailoring the task to the individual child) remains an important weakness of the SAM that should be addressed in the discussion. Notably, for administration, prompting by the experimenter arguably increases the engagement with the task, establishing the give-and-take nature of interactive play between the experimenter and the child. More importantly, for rating children who avoid a story-theme even after a standardized prompt are typically rated as higher in avoidance or denial, therefore providing evidence for disentangling attachment-avoidance and attachment-resistance. Moreover, this procedure is considered particularly important for young children so as to help distinguish potential lack of comprehension from avoidant manoeuvres.

3. I concur with the authors that readers of their manuscript will presumably mostly be psychologists and psychiatrists. However, I do not go along with their decision to therefore dispense with a balanced view on the limitations of the algorithm. In fact, I feel they are under an obligation to do so (in non-technical terms) in a separate paragraph in the discussion. For example, it is crucial to address whether their statistical leave-one-out cross validation approach may overestimate the generalisability of their algorithm (e.g., due to overfitting, sample size issues and a lack of a validation set) and I would venture that most statically well-trained psychologists will be able to grasp the gist of these issues and therefore be put in a better position to reach an informed decision as to whether they should use the SAM or not.

Finally, I do not understand why the authors cannot provide anonymised secondary data (e.g., spreadsheet codes derived from SAM vs. MCAST vs. the algorithm, age , gender) which could be useful for meta-analyses.

7. PLOS authors have the option to publish the peer review history of their article (what does this mean?). If published, this will include your full peer review and any attached files.

Reviewer #2: No

---

## [Author Response · Author response to Decision Letter 1]

16 Jun 2021

Dear Professor Taubner, We have followed your most recent advice and have added more detail on the machine learning algorithms in both Methods and limitations sections. We have used track changes to amend our response to reviewers. Thanks for all your help with this paper. Best wishes, Helen

---

## [Editor Report · Decision Letter 2]

5 Jul 2021

The School Attachment Monitor - a novel computational tool for assessment of attachment in middle childhood.

PONE-D-20-29493R2

Dear Dr. Minnis,

We’re pleased to inform you that your manuscript has been judged scientifically suitable for publication and will be formally accepted for publication once it meets all outstanding technical requirements.

Kind regards,

Svenja Taubner

Academic Editor

PLOS ONE
---

## [Editor Report · Acceptance letter]

12 Jul 2021

PONE-D-20-29493R2 

The School Attachment Monitor - a novel computational tool for assessment of attachment in middle childhood. 

Dear Dr. Minnis:

I'm pleased to inform you that your manuscript has been deemed suitable for publication in PLOS ONE. Congratulations! Your manuscript is now with our production department. 

Kind regards, 

on behalf of

Dr. Svenja Taubner 

Academic Editor

PLOS ONE